# Prevalence and Genetic Characterization of Morphologically Indistinguishable Sarcocysts of *Sarcocystis cruzi* in Cattle and *Sarcocystis poephagicanis* in Yaks

**Kui Tang [1,†], Danqu Lamu [2,†], Tao Qin [1], Zhe Liao [1], Mingzhu Zhang [1], Zhipeng Wu [1], Shuangsheng Deng [3], Jianping Tao [4] and Junjie Hu [1,*]**

1    Yunnan Key Laboratory for Plateau Mountain Ecology, Restoration of Degraded Environments, School of Ecology and Environmental Sciences, Yunnan University, Kunming 650091, China; ktang@mail.ynu.edu.cn (K.T.); qqt@mail.ynu.edu.cn (T.Q.); liaozhe@mail.ynu.edu.cn (Z.L.); mzzhang@mail.ynu.edu.cn (M.Z.); wzp@mail.ynu.edu.cn (Z.W.)
2    Institute of Animal Science, Tibet Academy of Agricultural and Animal Husbandry Sciences, Lhasa 850009, China; yyxf521369@163.com
3    School of Biological Sciences, Yunnan University, Kunming 650091, China; ssdeng@ynu.edu.cn
4    College of Veterinary Medicine, Yangzhou University, Yangzhou 225009, China; yzjptao@126.com
*    Correspondence: jjhu@ynu.edu.cn
†    These authors contributed equally to this work.

**Abstract:** *Sarcocystis cruzi* in cattle (*Bos taurus*) and *Sarcocystis poephagicanis* in yaks (*Bos grunniens*) are morphologically indistinguishable. However, the relationship between the two parasites is still unclear. Here, muscular tissues of the two species of domestic animal collected from abattoirs in China were examined for sarcocysts of *S. cruzi* and *S. poephagicanis*. The sarcocysts isolated from the samples were analyzed using light microscopy (LM), transmission electron microscopy (TEM), and DNA analysis. Sarcocysts of *S. cruzi* and *S. poephagicanis* were found in 405 of 950 (42.6%) cattle and 304 of 320 (95.0%) yaks. LM and TEM showed that the sarcocysts of the two parasites had similar morphological characteristics. The thin-walled sarcocysts had hair-like protrusions on the surface. The ultrastructures were demonstrated to include a primary cyst wall containing irregularly folded, hirsute, or bone-like protrusions. Four genetic markers of the two parasites were sequenced and analyzed, namely, *18S rDNA*, *28S rDNA*, mitochondrial *cox1*, and apicoplast *rpl6*. The sequences of the four loci had an interspecific similarity of 97.9–98.6%, 97.2–98.1%, 89.5–90.4%, and 96.9–97.2% identity, respectively. Phylogenetic analysis using *28S rDNA* and *cox1* sequences indicated that both *S. cruzi* and *S. poephagicanis* were placed into a group encompassing *Sarcocystis* spp. in ruminants with canid as known or putative definitive hosts. *Sarcocystis cruzi* and *S. poephagicanis* represent separate species, and *cox1* and *rpl6* were suitable for distinguishing between them.

**Keywords:** *Sracocystis cruzi*; *Sarcocystis poephagicanis*; cattle; yak; morphological and molecular characterization

## 1. Introduction

*Sarcocystis* spp. are cyst-forming intracellular protozoan parasites with an obligate two-host life cycle, with predators as the definitive hosts and their prey animals as the intermediate hosts. In the intermediate hosts, muscle sarcocysts are formed as the result of asexual reproduction, and in definitive hosts, sexual reproduction occurs in intestinal cells, with oocysts or sporocysts passed in feces. Collectively, these species have considerable veterinary, economic, and public health importance. At present, the classification and identification of *Sarcocystis* species in a given host mainly depends on the morphological characterization of its sarcocysts and nucleotide sequences of genetic markers [1].

*Sarcocystis cruzi* is found in cattle (*Bos taurus*) worldwide and is considered to be the most pathogenic *Sarcocystis* species in the animal [2]. *Sarcocystis poephagicanis* was

first found and described in yaks (*Bos grunniens*, synonym: *Poephagus grunniens*) [3], live-stock animals that are adapted to high-altitude environments and raised mainly on the Qinghai–Tibet Plateau, China, but also in adjacent areas [4]. The two parasites present similar morphological characteristics and life cycles (canids as definitive hosts) [2,3]. Owing to the unclear relationships between the two species, *S. poephagicanis* in yaks has frequently been identified as *S. cruzi* by some authors [5–7].

Currently, molecular analysis based on nucleotide sequences has been used to infer the phylogenetic relationship of *Sarcocystis* species and is proven to be a more useful and efficient tool for delineating or identifying *Sarcocystis* species than the traditional morphological method, especially for morphologically indistinguishable *Sarcocystis* spp. in different hosts [8,9]. Abundant nucleotide sequences of molecular markers, including *18S rDNA*, *28S rDNA*, and mitochondrial *cox1* genes of *S. cruzi*, have been deposited in GenBank as references. However, no DNA sequences of *Sarcocystis* spp. in yaks have been deposited in GenBank.

Therefore, the aims of the present study were (1) to investigate the prevalence of *S. cruzi* in cattle and *S. poephagicanis* in yaks in China based on morphological observation; (2) to analyze the molecular characteristics of four genetic markers of the two parasites, namely, *18S rDNA*, *28S rDNA*, *cox1*, and apicoplast large subunit ribosomal protein 6 (*rpl6*); and (3) to infer the phylogenetic relationships of the two species with other *Sarcocystis* spp. using *28S rDNA* and *cox1* sequences.

## 2. Materials and Methods

Muscular tissues obtained from 950 cattle and 320 yaks were separately collected from abattoirs in Kunming, the capital of Yunnan province, and Lhasa, the capital of Tibet autonomous region, both located in southwestern China, during 2021–2023. About 500 g of samples (esophagus, diaphragm, skeletal muscles, tongue, and heart) was obtained from each animal and shipped with dry ice to the zoological laboratory of Yunnan University. In the laboratory, approximately 40, 3 mm muscle pieces from each collected sample were compressed between two glass slides to detect the presence of sarcocysts using stereomicroscopy; individual sarcocysts were extracted and isolated from muscular fibers using dissection needles and processed using light microscopy (LM), transmission electron microscopy (TEM), and DNA analysis. For observing and measuring bradyzoites filled in sarcocysts, the sarcocysts were punctured using a needle. For TEM, the sarcocysts were fixed in 2.5% glutaraldehyde in cacodylate buffer (0.1 M, pH 7.4) at 4 °C and post-fixed in 1% osmium tetroxide in the same buffer, then dehydrated in graded alcohols and embedded in Epon-Alaldite mixture. Ultrathin sections were stained with uranyl acetate and lead citrate and then examined using a JEM100-CX TEM (JEOL Ltd., Tokyo, Japan). For DNA isolation, individual cysts were stored in sterile water at −20 °C prior to processing.

A total of 12 individual sarcocysts, including six of *S. cruzi* and six of *S. poephagicanis* isolated from cardiac muscles of different animals, were separately subjected to genomic DNA extraction using a TIANamp Genomic DNA Kit (Tiangen Biotech Ltd., Beijing, China) according to the manufacturer's instructions. Four genes, namely, *18S rDNA*, *28S rDNA*, *cox1*, and *rpl6*, were used to characterize the two parasites. The primer pairs used are given in Table 1. Polymerase chain reaction (PCR) amplifications were performed in a reagent mixture with a total volume of 25 μL that included 12.5 μL Green Taq Mix (Vazyme Biotech, Nanjing, China), 5.5 μL double-distilled $H_2O$, 1.0 μL of each primer (10 μM), and 5 μL template DNA. The cycling parameters slightly differed for each gene. For *18S rDNA*, the amplification reaction started with denaturation at 94 °C for 5 min, followed by 35 cycles of 94 °C for 1 min, 57 °C for 1 min, and 72 °C for 1.5 min, with a final extension at 72 °C for 10 min. For *28S rDNA, cox1*, and *rpl6*, the cycling parameters started with denaturation at 94 °C for 5 min, followed by 35 cycles of 94 °C for 1 min, 52 °C for 1 min, and 72 °C for 1.5 min, with a final extension at 72 °C for 7 min. The resulting PCR products were gel purified using an E.Z.N.A.® Gel Extraction Kit (Omega Bio-Tek, Inc., Norcross, GA, USA) and ligated to the pCE2 TA/Blunt-Zero vector using a 5 min TA/Blunt-Zero

Cloning Kit (Vazyme Biotech Co., Ltd., Nanjing, China) according to the manufacturer's instructions. The ligated vectors were transformed into Trelief® 5α Chemically Competent Cells (Tsingke Biotechnology Co., Ltd., Beijing, China). The selected positive bacterial clones were sequenced in both directions using an ABI PRISM TM 3730 XL DNA Analyzer (Applied Bio-systems, Thermo Fisher Scientific, Waltham, MA, USA).

**Table 1.** Primers used for the amplification of the four genes.

| Gene Name | Primer Name | Primer Sequence (5′–3′) | Reference |
|---|---|---|---|
| *18S rDNA* | ERIB1 [a] | ACCTGGTTGATCCTGCCAG | [10] |
| | B [b] | GATCCTTCTGCAGGTTCACCTAC | [11] |
| *28S rDNA* | KL1 [a] | TACCCGCTGAACTTAAGC | [12] |
| | KL3 [b] | CCACCAAGATCTGCACTAG | |
| | KL4 [a] | AGCAGGACGGTGGTC | |
| | KL5 [b] | CTCAAGCTCAACAGGGTC | |
| | KL6 [a] | GGATTGGCTCTGAGGG | |
| | KL2 [b] | ACTTAGAGGCGTTCAGTC | |
| *cox1* | SF1 [a] | ATGGCGTACAACAATCATAAAGAA | [13] |
| | SR9 [b] | ATATCCATACCRCCATTGCCCAT | [14] |
| *rpl6* | L6F [a] | CCATGAAACTTAATTTGCACA | This study |
| | L6R [b] | CTTAAAAGTTCTATTATGGGTT | |

[a] Forward primer; [b] reverse primer. The forward and reverse primers for *rpl6* used in this study were separately designed using OLIGO 5.0 (National BioScience, Plymouth, MN, USA) based on the apicoplast genomes of *Sarcocystis wenzeli* (unpublished data) and *Toxoplasma gondii* (NC001799).

Phylogenetic analyses were conducted separately on the nucleotide sequences of the *28S rDNA* and *cox1* sequences using MEGA X software Version 10.1.89 [15]. The evolutionary history was inferred using the minimum evolution (ME) method. The trees were drawn to scale, with branch lengths in the same units as those of the evolutionary distances used to infer the phylogenetic tree. The evolutionary distances were computed using the Kimura 2-parameter method and are in the units of the number of base substitutions per site. The percentage of replicate trees in which the associated taxa clustered together in the bootstrap test (1000 replicates) are shown next to the branches. All positions containing gaps and missing data were eliminated (complete deletion option).

Nucleotide sequences of *Sarcocystis* spp. used in the investigation were downloaded from GenBank. The *28S rDNA* and *cox1* sequences were aligned, respectively, using the ClustalW and MUSCLE program implemented in MEGA X. The alignments were subsequently checked visually; some sequences were slightly truncated at both ends so that all sequences started and ended at the same nucleotide positions, except that the *28S rDNA* sequence of *S. levinei* (KU247944) was shorter at the 3′ ends. The final *28S rDNA* alignment included 27 nucleotide sequences from 19 taxa and 3980 positions. The final *cox1* alignment included 33 nucleotide sequences from 25 taxa and 1014 positions. *Hammondia heydorni* and *Toxoplasma gondii* were chosen as outgroup species to root both trees.

### 3. Results

*3.1. Prevalence of S. cruzi in Cattle and S. poephagicanis in Yaks*

Sarcocysts of *S. cruzi* were found in 405 of 950 (42.6%) cattle, and sarcocysts of *S. poephagicanis* were found in 304 of 320 (95.0%) yaks with the aid of LM. Among the examined tissues, the highest prevalence of the two parasites was recorded in the hearts of the two animals, i.e., 40.5% for *S. cruzi* in cattle and 87.8% for *S. poephagicanis* in yaks (Table 2).

**Table 2.** Prevalence of *S. cruzi* in cattle and *S. poephagicanis* in yaks in China.

| Tissues Examined | *S. cruzi* | | | *S. poephagicanis* | | |
|---|---|---|---|---|---|---|
| | No. Sampled | No. Infected | % of Infected | No. Sampled | No. Infected | % of Infected |
| Esophagus | 301 | 110 | 36.5 | 284 | 227 | 79.9 |
| Tongue | 192 | 23 | 12.0 | 68 | 26 | 38.2 |
| Diaphragm | 947 | 298 | 31.5 | 286 | 232 | 81.1 |
| Heart | 780 | 316 | 40.5 | 279 | 245 | 87.8 |
| Skeletal muscles | 950 | 251 | 26.4 | 320 | 239 | 74.5 |
| Total infected animals | 950 | 405 | 42.6 | 320 | 304 | 95.0 |

### 3.2. LM and TEM of Sarcoysts of S. cruzi and S. poephagicanis

Using LM and TEM, the sarcocysts of *S. cruzi* and *S. poephagicanis* can be seen to have similar morphological characteristics (Figure 1). The sarcocysts were thin-walled and septate and had hair-like protrusions on the surface (Figure 1a,d). Sarcocysts of *S. cruzi* measured 256–1325 × 24–87 µm, and those of *S. poephagicanis* were 337–996 × 54–128 µm in size. The bradyzoites filled in the sarcocysts were banana-shaped, measuring 8.9–14.0 × 3.2–4.8 µm and 8.8–16.0 × 3.2–5.6 µm in size, respectively, for *S. cruzi* and *S. poephagicanis* (Figure 1b,e). Ultrastructurally, the primary cyst wall contained irregularly folded, hirsute, or bone-like protrusions (Figure 1c,f). A layer of ground substances measuring 0.6–0.8 µm in thickness was located immediately beneath the primary sarcocyst wall.

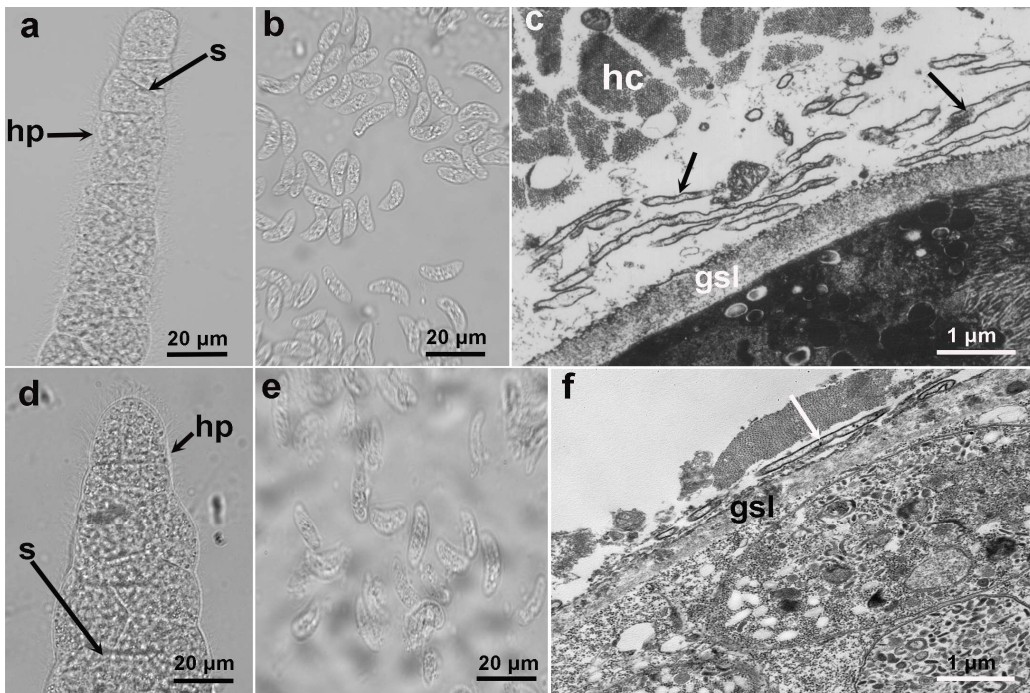

**Figure 1.** Morphological characteristics of sarcocysts of *Sarcocystis cruzi* and *Sarcocystis poephagicanis* obtained from cattle and domestic yaks, respectively. (**a**) *S. cruzi* sarcocyst had septae (s) and was surrounded by hair-like protrusions (hp) (unstained, light microscopy, LM). (**b**) Banana-shaped bradyzoites of *S. cruzi* (unstained, LM). (**c**) Diagonal section of a sarcocyst of *S. cruzi* (under transmission electron microscopy, TEM). Sarcocyst surrounded by host cell (hc), and hirsute or bone-like protrusions (arrow) presented on the surface of the ground substance layer (gsl). (**d**) *S. poephagicanis* sarcocyst. Note: septae (s) and hair-like protrusions (hp) (unstained, LM). (**e**) Bradyzoites of *S. poephagicanis*. (**f**) TEM of a *S. poephagicanis* sarcocyst. Note: hirsute or bone-like protrusions (arrow) and the ground substance layer (gsl).

### 3.3. Molecular Characterization of 18S rDNA, 28S rDNA, cox1, and rpl6

Genomic DNA was extracted from the individual sarcocysts of the two parasites isolated from different animals. The *18S rDNA*, *28S rDNA*, *cox1*, and *rpl6* were amplified successfully. Six nucleotide sequences of each gene for the two *Sarcocystis* species were analyzed in the present study. The *18S rDNA*, *28S rDNA*, *cox1*, and *rpl6* sequences of *S. cruzi* were 1857–1869 bp, 3464–3474 bp, 1085 bp, and 864 bp in length, and showed an intraspecific identity of 99.0–100% (on average 99.4%), 98.9–99.7% (on average 99.3%), 98.8–99.9% (on average 99.3%), and.99.7–100% (on average 99.8%), respectively. The *S. poephagicanis* sequences of the four genes were 1871–1873 bp, 3460–3469 bp, 1085 bp, and 864 bp long, and showed an intraspecific identity of 98.8–100% (on average 99.6%), 98.0–99.6% (on average 98.8%), 98.1–100% (on average 98.8%), and 99.7–100% (on average 99.8%), respectively. Meanwhile, the interspecific identities at the four loci were 97.9–98.6% (on average 98.3%), 97.2–98.1% (on average 97.7%), 89.5–90.4% (on average 89.9%), and 96.9–97.2% (on average 97.1%), respectively. The newly obtained sequences of the two parasites were deposited in GenBank under accession numbers OR553288–OR553292, OR573608–OR573623, OR570876–OR570884, and OR590796–OR590800.

When comparing the newly obtained sequences with those previously deposited in GenBank, at the four loci (*18S rDNA*, *28S rDNA*, *cox1*, and *rp16*), the sequences most similar to the *S. cruzi* sequences were those of *S. cruzi* (on average 99.3% identity), *S. cruzi* (on average 99.1% identity), *S. cruzi* (on average 99.1% identity), and *T. gondii* (72.7% identity), respectively, and the *S. poephagicanis* sequences had the highest similarity to those of *S. cruzi* (on average 98.7% identity), *S. levinei* (on average 97.5% identity) from water buffalo (*Bubalus bubalis*), *S. rangi* (on average 91.8% identity) from reindeer (*Rangifer tarandus*), and *T. gondii* (72.8% identity), respectively (Table 3).

**Table 3.** Similarities between the newly obtained sequences of *S. cruzi* and *S. poephagicanis* and those previously deposited in GenBank.

| Species | Genetic Markers | Accession Number | Comparison with Sequences Previously Deposited in GenBank | | |
| --- | --- | --- | --- | --- | --- |
| | | | Species | Accession Number | Identity% (Average%) |
| *S. cruzi* | *18S rDNA* | OR553288–OR553292 | *S. cruzi* | #1 | 98.7–99.8 (99.3) |
| | | | *S. levinei* | KU247914–KU247922 | 99.0–99.5 (99.2) |
| | | | *S. gjerdei* | LC481028–LC481031, LC349475–LC349479 | 98.1–98.6 (98.3). |
| | *28S rDNA* | OR573608–OR573613 | *S. cruzi* | KT901270–KT901285, AF076903 | 98.6–99.5 (99.1) |
| | | | *S. levinei* | KU247937–KU247945, MH793424–MH793426 | 98.1–98.6 (98.4) |
| | *cox1* | OR570876–OR57081 | *S. cruzi* | #2 | 96.4–99.8 (97.2) |
| | | | *S. levinei* | MH255771–MH255781, KU247874–KU247885 | 93.1–94.0 (93.6) |
| | *rpl6* | OR590796–OR590798 | *T. gondii* | NC001799 | 72.7 |
| *S. poephagicanis* | *18S rDNA* | OR573620–OR573623 | *S. cruzi* | #1 | 97.9–98.8 (98.7) |
| | | | *S. gjerdei* | LC481028–LC481031, LC349475–LC349479 | 98.2–98.7 (98.5) |
| | | | *S. levinei* | KU247914–KU247922 | 98.5 |
| | *28S rDNA* | OR573614–OR573619 | *S. levinei* | KU247937–KU247945, MH793424–MH793426 | 97.2–98.1 (97.5) |
| | | | *S. cruzi* | KT901270–KT901285, AF076903 | 95.2–98.1 (97.2) |

**Table 3.** *Cont.*

| Species | Genetic Markers | Accession Number | Comparison with Sequences Previously Deposited in GenBank | | |
|---|---|---|---|---|---|
| | | | Species | Accession Number | Identity% (Average%) |
| *S. poephagicanis* | *cox1* | OR570882–OR570884 | *S. rangi* | KC209662–KC209668 | 91.6–91.9 (91.8) |
| | | | *S. cruzi* | #2 | 89.4–90.5 (90.0) |
| | | | *S. levinei* | MH255771–MH255781, KU247874–KU247885 | 88.8–89.3 (89.1) |
| | *rpl6* | OR590799–OR590800 | *T. gondii* | NC001799 | 72.8 |

#1 KT901167, JX679467, JX679468, LC171827–LC171830, KC209738, AB682779, AB682780, and AF017120;
#2 MK962349–MK962351, LC171859–LC171862, KC209597–KC209600, KT901078–KT901095, MT796926–MT796945, MW507158, MW507159, MW490605, MW490606, and MG787071–MG787076.

### 3.4. Phylogenetic Analysis

Phylogenetic analysis based on the *28S rDNA* and *cox1* sequences of S. *cruzi* and *S. poephagicanis* showed a similar tree topology (Figure 2), and they were placed into a group encompassing *Sarcocystis* spp. in ruminants with canids as known or putative definitive hosts. In the tree inferred from *28S rDNA* (Figure 2a), *S. poephagicanis* newly sequenced isolates formed an individual clade clustered with a clade formed by *S. cruzi* and *S. levinei*. In the tree inferred from *cox1* (Figure 2b), *S. poephagicanis* clustered with *S. rangi*, *S. hjoti* from red deer (*Cervus elaphus*), *S. gjerdei* from sika deer (*C. nippon*), and *S. alceslatrans* from Canadian moose (*Alces alces*), which separated from the clade formed by *S. cruzi* and *S. levinei*.

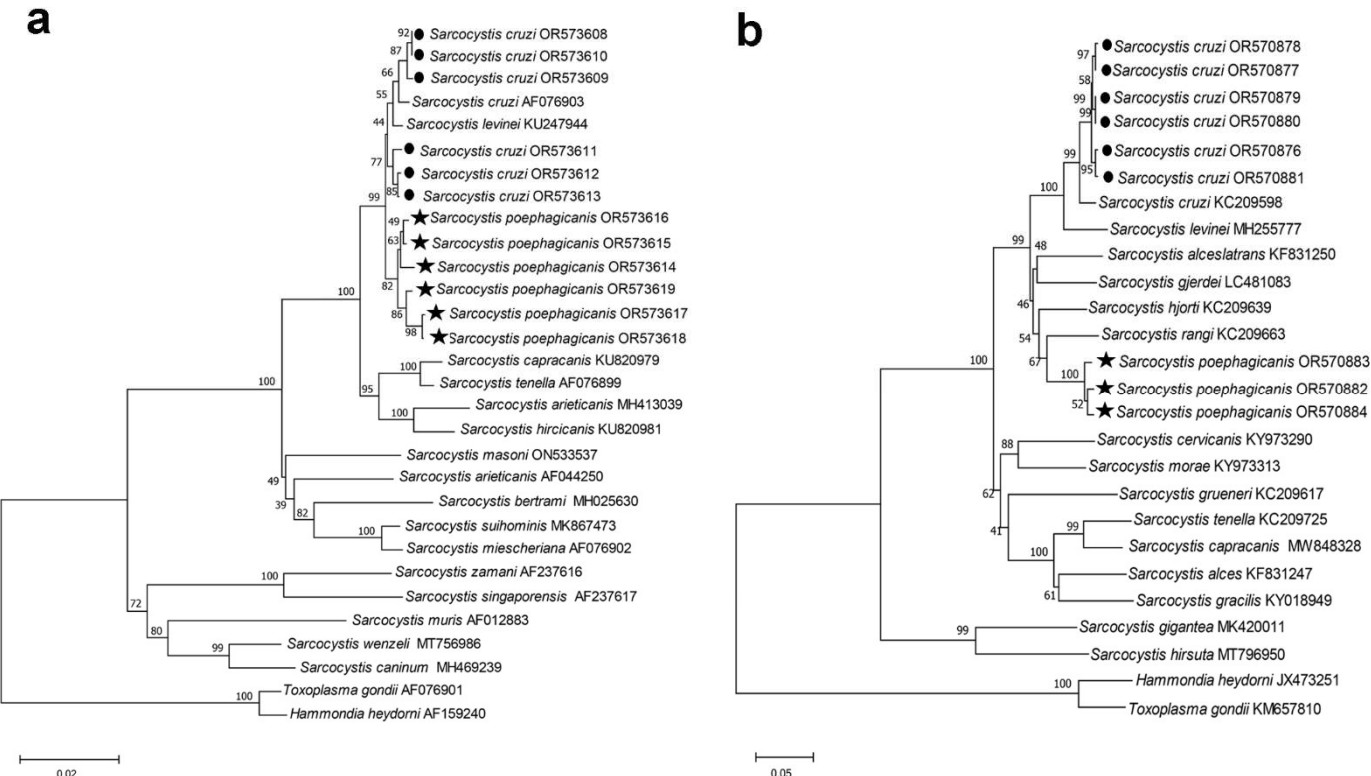

**Figure 2.** Phylogenetic trees of selected *Sarcocystis* species. The trees are constructed based on (**a**) *28S rDNA* sequences and (**b**) mitochondrial *cox1* sequences using the minimum evolution (ME) method with the Kimura 2-parameter method computing the evolutionary distances. The percentage of replicate trees in which the associated taxa clustered together in the bootstrap test (1000 replicates) are shown next to the branches. The trees are drawn to scale, with branch lengths in the same units as

those of the evolutionary distances used to infer the phylogenetic tree. (**a**) The six newly obtained *28S rDNA* sequences of *S. cruzi* (OR573608–OR573613, highlighted by circular symbols) formed a clade with *S. cruzi* and *S. levinei*, which clustered with a clade formed by six newly sequenced *S. poephagicanis* isolates (OR57614–OR573619, highlighted by pentagonal symbols). (**b**) The six newly obtained *cox1* sequences of *S. cruzi* (OR570876–OR57081, highlighted by circular symbols) formed a clade with *S. cruzi* and *S. levinei*, which separated from a clade formed by the three newly obtained *S. poephagicanis cox1* sequences (OR57082–OR57084, highlighted by pentagonal symbols) and *S. rangi*. Both clades were part of the group *Sarcocystis* spp. in ruminants with canids as known or putative definitive hosts.

## 4. Discussion

At present, at least seven *Sarcocystis* species are known to exist in cattle, namely, *S. cruzi*, *S. heydorni*, *S. bovini*, *S. hirsuta*, *S. rommeli*, *S. hominis*, and *S. bovifelis*. These species could be divided into two categories based on the thickness of the sarcocyst walls. Sarcocysts of *S. cruzi* and *S. heydorni* are thin-walled, and those of the remaining five species are thick-walled [1,2]. Although there is some confusion concerning the relationship of thick-walled species of *Sarcocystis* in cattle [2,8,9], *S. cruzi* is the indisputable member of this group due to its unique morphological features: thin-walled sarcocysts and the surface of its cyst walls being covered with hair-like (under LM) or ribbon-like (under TEM) protrusions [2]. Two *Sarcocystis* species were discovered and named in yaks, i.e., *S. poephagicanis* and *S. poephagi* [3]. To date, there are only two references [3,16] providing the morphological characteristics of the two parasites. In the original description, sarcocysts of *S. poephagicanis* are microscopic and thin-walled, and those of *S. poephagi* are macroscopic and thick-walled under LM. However, the ultrastructural characteristics of the sarcocysts of the two parasites were not accurately detailed. According to the figures provided by Wei et al. [3,16], we can observe the primary cyst wall of *S. poephaicanis* covered with short ribbon-like protrusions and that of *S. poephagi* covered with closely packed long villar protrusions, which are similar to type 7a and type 18, respectively, according to the TEM cyst wall type classified by Dubey et al. [1]. Probably owing to the limitation of the original description for *S. poephagicanis* and its high morphological similarities with *S. cruzi* in cattle, the thin-walled sarcocysts in yaks were frequently thought to be *S. cruzi* in the epidemiological surveillance of sarcocystosis (mentioned in the Section 1).

*Sarcocystis cruzi* has been diagnosed in cattle throughout the world, with its prevalence rate ranging from 29.6% to 100% [1]. To date, almost all accounts of *Sarcocystis* spp. in yaks have been reported in China, with the prevalence of sarcocysts in yaks ranging from 14.7% to 100% [3,5–7,16]. Here, with the aid of LM, the prevalence rate of *S. cruzi* in Chinese cattle was 42.6%, lower than 95.0% for *S. poephagicanis* in Chinese yaks. The difference in the prevalence rate of the two species may be due to the more intensive culture of cattle, as the free-range farming of yaks is still popular in China, meaning that yaks are more likely than cattle to encounter the feces of domestic dogs.

Molecular markers have been extensively used to identify *Sarcocystis* spp. in different animals, and different genes have presented different discriminative abilities. For example, *18S rDNA* has been proven to be unsuitable for distinguishing between the closely related *Sarcocystis* spp. in the same or different ruminant animals, and *cox1* has been proven more suitable for distinguishing the closely related species of *Sarcocystis* [13,14]. In the present study, the four genetic markers, namely, *18S rDNA*, *28 rDNA*, *cox1*, and *rpl6*, of the two parasites were sequenced and analyzed. The interspecific similarities at the four loci were 97.9–98.6%, 97.2–98.1%, 89.5–90.4%, and 96.9–97.2% identity, respectively, which indicated that the four genes could be used to distinguish between them; however, *cox1* and *rpl6* were more suitable for this.

Phylogenetic analysis based on *28S rDNA* and *cox1* indicated that *S. poephagicanis* and *S. crzui* were part of the group *Sarcocystis* spp. in ruminants with canids as defini-

tive hosts. Meanwhile, *S. cruzi* and *S. levinei* formed an individual clade separate from *S. poephagicanis*, and it was revealed that *S. cruzi* had a closer relationship with *S. levinei* than *S. poephagicanis.* Sarcocysts of *S. levinei* are morphologically undistinguishable from *S. cruzi* and *S. poephagicanis* [1,3]. The relationship between *S. cruzi* and *S. levine* has thus been determined, and based on the divergence of their cox1 sequences, they supposedly represent separate species with different hosts [17].

## 5. Conclusions

In the present study, sarcocysts of *S. cruzi* in cattle and *S. poephagicanis* in yaks were detailed morphologically. Additionally, the four genetic markers of the two parasites were sequenced and analyzed, and the sequences of *S. poephagicanis* constitute the first records of *Sarcocystis* spp. from yaks in GenBank. Based on molecular analysis, the two morphologically indistinguishable species represent separate species in different hosts.

**Author Contributions:** Conceptualization, J.H.; methodology, K.T.; software, T.Q.; validation, S.D. and D.L.; formal analysis, Z.L., M.Z. and Z.W.; investigation, K.T. and D.L.; resources, K.T. and D.L.; data curation, K.T. and D.L.; writing—original draft preparation, J.H.; writing—review and editing, J.H. and J.T.; supervision, J.H.; project administration, J.H.; funding acquisition, J.H. All authors have read and agreed to the published version of the manuscript.

**Funding:** This research was funded by the Natural Science Foundation of China, grant number 31460557, and the Research Project of Tibet Autonomous Region, grant number XZ202301ZY0006N.

**Institutional Review Board Statement:** The animal study protocol was approved by the Animal Ethics Committee of Yunnan University (permission number AECYU2018004).

**Data Availability Statement:** Data is contained within the article.

**Conflicts of Interest:** The authors declare no conflict of interest.

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
