# Peer review of "Prevalence and Genetic Characterization of Morphologically Indistinguishable Sarcocysts of Sarcocystis cruzi in Cattle and Sarcocystis poephagicanis in Yaks"

_diversity, doi:10.3390/d15111136_

Round 1
Reviewer 1 Report
Comments and Suggestions for Authors
The authors studied Sarcocystis parasites in cattle and yaks in Tibet and characterized Sarcocystis poephagicanis genetically by analyzing two nuclear genes (18S rRNA and 28S rRNA genes), the mitochondrial cox1 gene, and the apicoplast rpl6 gene in several samples. They investigated tissue samples of 950 cattle and 320 yaks for the presence of sarcocysts and found high prevalences of S. cruzi in cattle and S. poephagicanis.
I recommend the study for publication. However, the manuscript should be carefully checked for grammatical errors.
They should show the trees as phylograms, not as cladograms. Phylograms are better for representing phylogenies because they show the branch lengths.
All scientific taxon names and gene symbols (18S, 28S, cox1, and rpl6) should be written in italics.
Abstract:
Line 3: I would change to "the two domestic animals" to
"the two domestic animal species", otherwise it sounds like only two individuals were examined.
L 10: Change all gene symbols to italics: 18S, 28S, cox1, and rpl6.
L 12: "The sequences of the four loci presented an interspecific identity of ...". I would change "interspecific identity" to "interspecific similarity of ..."
L 14: Change "with canid as" to "with canids as".
L 15: Change "cox1 and rpl6 was suitable" to "cox1 and rpl6 were suitable".
Introduction:
Page 1:
Change "Sarcocystis cruzi in cattle (Bos taurus) distributes worldwide" to "Sarcocystis cruzi in cattle (Bos taurus) is distributed worldwide".
Change "Sarcocystis poephagicanis is described and named in yaks (Bos grunniens)" to "Sarcocystis poephagicanis was first found and described in yaks (Bos grunniens; synonym: Poephagus grunniens)".
Wei et al. (1985) in the title refer to the yak as "Poeophagus grunniens". However, the correct name of the synonym should be Poephagus grunniens without "o". Maybe a typo in the references.
Page 2:
Introduction:
Change "Owing to the relationship between them unclear, ..." to "Owing to the unclear relationships between the two species, ...".
Change "and recommended to be a more useful and efficient tool" to "and proved to be a more useful and efficient tool".
Change "cox1 genes, of S. cruzi" to "cox1 genes of S. cruzi".
Change "However, none of nucleotide sequences of Sarcocystis spp. in yaks have been investigated and provided in GenBank." to "However, no DNA sequences of Sarcocystis spp. in yaks have been deposited in GenBank."
Material and Methods:
Change "shipped with dry ice to zoological laboratory of Yunnan university" to "shipped with dry ice to the zoological laboratory of the Yunnan University".
Remove the comma here "approximately 40, 3 mm".
What do you mean with "80 kV" in "using a JEM100-CX TEM (JEOL Ltd., Tokyo, Japan) at 80 kV"?
Change "The primer pairs used were given in Table 1." to "The primer pairs used are given in Table 1."
Change "positive bacterial clones were sequenced on both directions" to "positive bacterial clones were sequenced in both directions".
Page 3:
Please indicate the annealing temperatures used in the PCRs in Table 1.
Change "The final alignment of the 28S rDNA sequences consisted of 27 nucleotide sequences and 3980 positions including gaps from 16 taxa." to "The final 28S rDNA alignment included 27 nucleotide sequences from 16 taxa and 3980 positions including gaps."
Results
Change S. cruzi and S. poephagicanis to italics throughout the text.
Page 4:
Change "The 18S rDNA, 28S rDNA, cox1, and rpl6 were amplified successfully using their DNAs as templates." to "The 18S rDNA, 28S rDNA, cox1, and rpl6 were amplified successfully."
Change "and shared an intraspecific identity of" to "and showed an intraspecific similarity of". Also in the next sentence.
Change "Meanwhile, at the four loci, the interspecific identity was ..." to "Meanwhile, the interspecific similarities at the four loci were ..."
Remove the comma here "under accession numbers, ...".
Page 5:
Again, change the taxon names Sarcocystis cruzi and Sarcocystis poephagicanis to italics in the legend of Figure 1 and elsewhere.
Page 6:
Show the trees as phylograms, not as cladograms. Phylograms are better for representing phylogenies because they also show the branch lengths.
Page 7:
Discussion
Change "Presntly, at least seven Sarcocystis species are recorded in cattle, namely S. cruzi ..." to "Presently, at least seven Sarcocystis species are known in cattle, namely S. cruzi ...".
Change "Two Sarcocystis species are discovered and named in yaks" to "Two Sarcocystis species were discovered and named in yaks".
Change "... two refereces [3, 18] provided morphological ..." to "... two references [3, 18] providing morphological ...".
Change "In the presentstudy, the four genetic markers" to "In the present study, the four genetic markers".
Change "and its high morphologically simialrities with S. cruzi in cattle" to "and its high morphological similarities with S. cruzi in cattle".
Change "the world, and prevalence rate of its sarcocysts ranged" to "the world, and prevalence rates ranged".
Change "and different genetic genes have presented different discriminative abilities" to "and different genes have presented different discriminative abilities".
Change "and cox1 has been recommended more suitable for distingushing the closely related speices of Sarcocystis [15]." to "and cox1 has been proven more suitable for distinguishing the closely related species of Sarcocystis [15]."
Change "At the four loci, the similarities between them were 97.9−98.6%, 97.2−98.1%, 89.5−90.4%, and 96.9−97.2% identity, respectively, which indiacted that the four genes could distingush them, and the cox1 and rpl6 were more suitable." to "At the four loci, the similarities were 97.9−98.6%, 97.2−98.1%, 89.5−90.4%, and 96.9−97.2%, respectively, which indicates that the four genes could distinguish them but cox1 and rpl6 were more suitable."
Change "Sarcocystis spp. in ruminants with canids definitive hosts." to "Sarcocystis spp. in ruminants with canids as definitive hosts."
Change "The relationship between S. cruzi and S. levine has been resolved, and based on the divergence of their cox1 sequnences, they were supposed to repersent separated species in different hosts [21]." to "The relationship between S. cruzi and S. levine has been resolved, and based on the divergence of their cox1 sequences, they supposedly represent separate species with different hosts [21]."
Comments on the Quality of English LanguageAll comments in the above section
Author Response
Thanks a lot for your excellent and responsible review for our manucript. Followings are our response to your kindly suggestions.
Abstract:
Line 3: I would change to "the two domestic animals" to "the two domestic animal species", otherwise it sounds like only two individuals were examined.
Response: the sentence has been edited as “ the two species of domestic animal”
L 10: Change all gene symbols to italics: 18S, 28S, cox1, and rpl6.
Response: all gene symbols presented in the manuscript has been changed to italics.
L 12: "The sequences of the four loci presented an interspecific identity of ...". I would change "interspecific identity" to "interspecific similarity of ..."
Response: accepted.
L 14: Change "with canid as" to "with canids as".
Response: accepted.
L 15: Change "cox1 and rpl6 was suitable" to "cox1 and rpl6 were suitable".
Response: accepted.
Introduction:
Page 1:
Change "Sarcocystis cruzi in cattle (Bos taurus) distributes worldwide" to "Sarcocystis cruzi in cattle (Bos taurus) is distributed worldwide".
Response: the sentence has been edited as “ Sarcocystis cruzi is found in cattle (Bos taurus) worldwide”
Change "Sarcocystis poephagicanis is described and named in yaks (Bos grunniens)" to "Sarcocystis poephagicanis was first found and described in yaks (Bos grunniens; synonym: Poephagus grunniens)".
Response: accepted.
Wei et al. (1985) in the title refer to the yak as "Poeophagus grunniens". However, the correct name of the synonym should be Poephagus grunniens without "o". Maybe a typo in the references.
Response: The spelling-mistake was corrected in the reference.
Page 2:
Introduction:
Change "Owing to the relationship between them unclear, ..." to "Owing to the unclear relationships between the two species, ...".
Response: accepted
Change "and recommended to be a more useful and efficient tool" to "and proved to be a more useful and efficient tool".
Response: accepted
Change "cox1 genes, of S. cruzi" to "cox1 genes of S. cruzi"
Response: accepted
Change "However, none of nucleotide sequences of Sarcocystis spp. in yaks have been investigated and provided in GenBank." to "However, no DNA sequences of Sarcocystis spp. in yaks have been deposited in GenBank."
Response: accepted
Material and Methods:
Change "shipped with dry ice to zoological laboratory of Yunnan university" to "shipped with dry ice to the zoological laboratory of the Yunnan University".
Response: accepted
Remove the comma here "approximately 40, 3 mm".
Response: accepted
What do you mean with "80 kV" in "using a JEM100-CX TEM (JEOL Ltd., Tokyo, Japan) at 80 kV"?
Response: 80 KV means eletron-beam accelerate voltage. Owing to this parameter is conventional, in the newly edited manuscript, at 80 kv has been deleted.
Change "The primer pairs used were given in Table 1." to "The primer pairs used are given in Table 1."
Response: accepted
Change "positive bacterial clones were sequenced on both directions" to "positive bacterial clones were sequenced in both directions".
Response: accepted
Page 3:
Please indicate the annealing temperatures used in the PCRs in Table 1.
Response: accepted
Change "The final alignment of the 28S rDNA sequences consisted of 27 nucleotide sequences and 3980 positions including gaps from 16 taxa." to "The final 28S rDNA alignment included 27 nucleotide sequences from 16 taxa and 3980 positions including gaps."
Response: accepted
Results
Change S. cruzi and S. poephagicanis to italics throughout the text.
Response: accepted. The species’ names have been checked throughout the manuscript again.
Page 4:
Change "The 18S rDNA, 28S rDNA, cox1, and rpl6 were amplified successfully using their DNAs as templates." to "The 18S rDNA, 28S rDNA, cox1, and rpl6 were amplified successfully."
Response: accepted.
Change "and shared an intraspecific identity of" to "and showed an intraspecific similarity of". Also in the next sentence.
Response: accepted.
Change "Meanwhile, at the four loci, the interspecific identity was ..." to "Meanwhile, the interspecific similarities at the four loci were ..."
Response: accepted.
Remove the comma here "under accession numbers, ...".
Response: accepted. The comma has been deleted.
Page 5:
Again, change the taxon names Sarcocystis cruzi and Sarcocystis poephagicanis to italics in the legend of Figure 1 and elsewhere.
Response: checked and corrected.
Page 6:
Show the trees as phylograms, not as cladograms. Phylograms are better for representing phylogenies because they also show the branch lengths.
Response: In the newly editerd manuscript, we also select cladograms to reflect the relationships of the two species. For the reasons that many published papers presently use cladograms to process phylogenetic analysis. After I received the suggestion, I attempted to reconstruct the phylogenetic trees with ME method using Mega X. This method can show the branch length, but the best DNA/Protein models calculated by the program in the software could not be selected while using ME. From my view, the phylogenetic trees only roughly reveal the relationships of different organism, probably I am not very good at it, and the methods mainly borrow from other papers.
Page 7:
Discussion
Change "Presntly, at least seven Sarcocystis species are recorded in cattle, namely S. cruzi ..." to "Presently, at least seven Sarcocystis species are known in cattle, namely S. cruzi ...".
Response: accepted.
Change "Two Sarcocystis species are discovered and named in yaks" to "Two Sarcocystis species were discovered and named in yaks".
Response: accepted.
Change "... two refereces [3, 18] provided morphological ..." to "... two references [3, 18] providing morphological ...".
Response: accepted.
Change "In the presentstudy, the four genetic markers" to "In the present study, the four genetic markers".
Response: accepted.
Change "and its high morphologically simialrities with S. cruzi in cattle" to "and its high morphological similarities with S. cruzi in cattle".
Response: accepted.
Change "the world, and prevalence rate of its sarcocysts ranged" to "the world, and prevalence rates ranged".
Response: the sentence has been edited as “Sarcocystis cruzi has been diagnosed in cattle throughout the world, with its prevalence rate ranging from 29.6% to 100% [1]”.
Change "and different genetic genes have presented different discriminative abilities" to "and different genes have presented different discriminative abilities".
Response: accepted
Change "and cox1 has been recommended more suitable for distingushing the closely related speices of Sarcocystis [15]." to "and cox1 has been proven more suitable for distinguishing the closely related species of Sarcocystis [15]."
Response: accepted
Change "At the four loci, the similarities between them were 97.9−98.6%, 97.2−98.1%, 89.5−90.4%, and 96.9−97.2% identity, respectively, which indiacted that the four genes could distingush them, and the cox1 and rpl6 were more suitable." to "At the four loci, the similarities were 97.9−98.6%, 97.2−98.1%, 89.5−90.4%, and 96.9−97.2%, respectively, which indicates that the four genes could distinguish them but cox1 and rpl6 were more suitable."
Response: the sentence has been edited as “The interspecific similarities at the four loci were 97.9−98.6%, 97.2−98.1%, 89.5−90.4%, and 96.9−97.2% identity, respectively, which indicated that the four genes could be used to distinguish between them; however, cox1 and rpl6 were more suitable for this.”
Change "Sarcocystis spp. in ruminants with canids definitive hosts." to "Sarcocystis spp. in ruminants with canids as definitive hosts."
Response: accepted.
Change "The relationship between S. cruzi and S. levine has been resolved, and based on the divergence of their cox1 sequnences, they were supposed to repersent separated species in different hosts [21]." to "The relationship between S. cruzi and S. levine has been resolved, and based on the divergence of their cox1 sequences, they supposedly represent separate species with different hosts [21]."
Response: the sentence has been edited as “The relationship between S. cruzi and S. levine has thus been determined, and based on the divergence of their cox1 sequences, they supposedly represent separate species with different hosts”.
Thank you again
Reviewer 2 Report
Comments and Suggestions for Authors
The aims of present study were to investigate the prevalence of S. cruzi in cattle and S. poephagicanis in yaks in China.
I think it's interesting but it's not a discovery but a study.
It is a genetics study therefore it could be useful to those who work in the sector in that specific area .
For the type of work setup, in my opinion the authors do not need to change anything.
Author Response
The aims of present study were to investigate the prevalence of S. cruzi in cattle and S. poephagicanis in yaks in China.
I think it's interesting but it's not a discovery but a study.
It is a genetics study therefore it could be useful to those who work in the sector in that specific area .
For the type of work setup, in my opinion the authors do not need to change anything.
Response: thanks a lot for your reviewe and kindly suggestions.
Reviewer 3 Report
Comments and Suggestions for Authors
The objectives of the were to investigate the prevalence of Sarcocystis in cattle and yak, to describe the characteristics of four genetic markers and to assess phylogenetic relationships of the two species.
This is an interesting study, which can become acceptable for publication after making the necessary changes, as indicated herebelow.
Major issues.
-Was there any strategy in collecting in tissue samples from the abattoirs? Please describe in brief and provide some details.
-How did you chose to study the characteristics of these 12 individual sarcocyctes out of the >700 that were obtained in the first place from the tissue samples?
-Can you please indicate the specific tissues from which these 12 individual sarcocyctes were recovered?
Minor issues.
-Please present all the details of the PCR, not just the primers.
-Table 2: please include some graphs with these findings to make these results more easy for readers to grasp.
-The phylogenetic trees must be designed again to be more clear, please.
Overall.
As I mentioned, this is an interesting study. The authors should very careful in addressing the above comments and should produce a revised version with appropriate corrections to continue the evaluation before possible acceptance.
Author Response
The objectives of the were to investigate the prevalence of Sarcocystis in cattle and yak, to describe the characteristics of four genetic markers and to assess phylogenetic relationships of the two species.
Thanks a lot for your kindly suggestions. The following is our response to the suggestions.
Major issues.
-Was there any strategy in collecting in tissue samples from the abattoirs? Please describe in brief and provide some details.
Response: There are no special strategies for collecting samples. We contacted the bosses of the abattoirs, bought the samples at the prices higher than markets. The bosses assigned the employee to chip off pieces of meat including different body parts. The different body parts were separately placed into small sealed plastic bags and labeled using marker pen. Samples collected from one animal were put into a big bag. The samples were stored in refrigerating cabinet. Most of samples were shipped to our laboratory but some samples were taken back by the postgraduates. I consider the progress of sampling is very common. Therefore, the detail of the process were not mentioned in the manuscript.
-How did you chose to study the characteristics of these 12 individual sarcocyctes out of the >700 that were obtained in the first place from the tissue samples?
Response: The sarcocysts used for TEM and molecular analysis in the present study were randomly chosen.
-Can you please indicate the specific tissues from which these 12 individual sarcocyctes were recovered?
Response: All sarcocysts used for molecualr analysis were isoalted from cardiac muscles of two species of two aniamls, owing to the sarcocysts were more easily separated from heart using needles comparing with other tissues. In the newly edited manuscript, the sentence was edited as " A total of 12 individual sarcocysts, including six of S. cruzi and six of S. poephagicanis isolated from cardiac muscles of different animals,"
Minor issues.
-Please present all the details of the PCR, not just the primers.
Response: The PCR process was not detailed for its conventional in the newly edited manuscript. However we added the annealing temperatures for each pair primers in Table 1.
Table 2: please include some graphs with these findings to make these results more easy for readers to grasp.
-The phylogenetic trees must be designed again to be more clear, please.
The phylogenetic trees must be designed again to be more clear, please.
Respons: Thee phylogenetic trees have been designed again. The newly obtained sequences of the two species were marked with two symbols. We hope they can more clear.
Overall.
As I mentioned, this is an interesting study. The authors should very careful in addressing the above comments and should produce a revised version with appropriate corrections to continue the evaluation before possible acceptance.
Thanks again.
Round 2
Reviewer 3 Report
Comments and Suggestions for Authors
The authors have improved the manuscript and I have no more comments.
Author Response
thank you